# The Mechanism of Kelulut Honey in Reversing Metabolic Changes in Rats Fed with High-Carbohydrate High-Fat Diet

**DOI:** 10.3390/molecules28062790

**Published:** 2023-03-20

**Authors:** Khairun-Nisa Hashim, Kok-Yong Chin, Fairus Ahmad

**Affiliations:** 1Department of Anatomy, Faculty of Medicine, Universiti Kebangsaan Malaysia, Jalan Yaacob Latif, Bandar Tun Razak, Kuala Lumpur 56000, Malaysia; 2Department of Pharmacology, Faculty of Medicine, Universiti Kebangsaan Malaysia, Jalan Yaacob Latif, Bandar Tun Razak, Kuala Lumpur 56000, Malaysia; chinkokyong@ppukm.ukm.edu.my

**Keywords:** metabolic syndrome, honey, obesity, hyperglycaemia, hyperlipidaemia, hypertension

## Abstract

Metabolic syndrome (MetS) is composed of central obesity, hyperglycemia, dyslipidemia and hypertension that increase an individual’s tendency to develop type 2 diabetes mellitus and cardiovascular diseases. Kelulut honey (KH) produced by stingless bee species has a rich phenolic profile. Recent studies have demonstrated that KH could suppress components of MetS, but its mechanisms of action are unknown. A total of 18 male Wistar rats were randomly divided into control rats (C group) (*n* = 6), MetS rats fed with a high carbohydrate high fat (HCHF) diet (HCHF group) (*n* = 6), and MetS rats fed with HCHF diet and treated with KH (HCHF + KH group) (*n* = 6). The HCHF + KH group received 1.0 g/kg/day KH via oral gavage from week 9 to 16 after HCHF diet initiation. Compared to the C group, the MetS group experienced a significant increase in body weight, body mass index, systolic (SBP) and diastolic blood pressure (DBP), serum triglyceride (TG) and leptin, as well as the area and perimeter of adipocyte cells at the end of the study. The MetS group also experienced a significant decrease in serum HDL levels versus the C group. KH supplementation reversed the changes in serum TG, HDL, leptin, adiponectin and corticosterone levels, SBP, DBP, as well as adipose tissue 11β-hydroxysteroid dehydrogenase type 1 (11βHSD1) level, area and perimeter at the end of the study. In addition, histological observations also showed that KH administration reduced fat deposition within hepatocytes, and prevented deterioration of pancreatic islet and renal glomerulus. In conclusion, KH is effective in preventing MetS by suppressing leptin, corticosterone and 11βHSD1 levels while elevating adiponectin levels.

## 1. Introduction

Metabolic syndrome (MetS) is composed of five clusters of risk factors which include central obesity, hyperglycemia, dyslipidemia and hypertension that increase an individual’s susceptibility to type 2 diabetes mellitus (T2DM) and cardiovascular disease (CVD) [1]. According to the International Diabetes Federation (IDF) in 2006, an estimated 20–25% of the adult population worldwide suffers from MetS [2]. While in another study in 2017, around 12–37% of the Asian population and 12–26% of the European population suffered from MetS [3].

Increased consumption of high-calorie foods and reduced physical activity trigger the formation of adipose tissue in the internal or visceral organs [4]. The accumulation of adipose tissue causes tissue hypoxia, which leads to the release of free fatty acids (FFA) and adipokines by adipose tissue, contributing to the complications of MetS [5]. The release of FFA and cytokines, such as tumour necrosis factor-alpha (TNF-α) and interleukin-1-beta (IL-1β), can reduce the sensitivity of insulin function in insulin-sensitive tissues, leading to insulin resistance [6]. The inability of these tissues to correct insulin resistance causes hyperglycemia and increases the risk of developing T2DM [4]. In addition, the release of FFA into the blood through splanchnic circulation into the liver tissue stimulates the breakdown of FFA for the production of TG and VLDL [7]. The production of excessive VLDL particles is closely related to the renal clearance of HDL particles [8]. Meanwhile, increased leptin and reduced adiponectin secretions by adipose tissue also play a role in causing MetS risk factors [4]. Studies have shown that the hormone leptin is closely related to increased renal sympathetic activity and blood pressure [9,10]. Meanwhile, the secretion of adiponectin, which is an important hormone in increasing insulin sensitivity, is reduced in obese individuals [11]. Reduced adiponectin secretion is associated with insulin resistance, which can increase the risk of MetS [12,13].

Lifestyle and dietary modifications continue to be the major preventative approach for metabolic syndrome [14]. However, pharmacological therapies are commonly used to treat each of the MetS risk factors [15]. The use of functional food as an alternative approach for MetS prevention is actively being investigated nowadays. The presence of multiple bioactive compounds in functional food can target multiple pathways contributing to the development of MetS [16]. Bee-derived products or apitherapy rich in polyphenol content could be effective in managing MetS [17]. Several studies have found that bee honey can inhibit MetS by acting as an anti-inflammatory, anti-obesity and anti-hypertensive agent [18,19,20].

Kelulut honey (KH) is a type of stingless bee honey found in Malaysia [21]. KH is known to have a high content of antioxidants in comparison to other local honeys [22]. Moreover, studies show that KH can suppress each component of MetS through its antioxidant and anti-inflammatory effects [23,24]. However, the mechanisms of action of honey in managing metabolic syndrome remain elusive. Therefore, this study was conducted to study the mechanism of KH in inhibiting MetS in rats given a high-carbohydrate high-fat (HCHF) diet.

## 2. Results

### 2.1. Liquid Chromatography-Mass Spectrometry (LC-MS) Analysis of Kelulut Honey

Profiling of KH through LC-MS found that KH contains various phenolic compounds consisting of phenolic acid and flavonoid groups (Table 1). These phenolic compounds may contribute to KH’s ability to inhibit metabolic changes in rats receiving the HCHF diet.

### 2.2. Changes in Weight, Waist Circumference, BMI and Fat Percentage

In this study, the HCHF diet caused an increase in obesity parameters such as weight (*p* < 0.001), fat percentage (*p* < 0.001), BMI (*p* < 0.001) and abdominal circumference (*p* = 0.006) in the HCHF group compared to the C group after 8 weeks. However, at week 16, only body weight (*p* < 0.001) and BMI (*p* < 0.001) were significantly increased compared to the C group. Meanwhile, KH supplementation for 8 weeks did not significantly reduce any of the obesity parameters in the HCHF + KH group significantly compared to the HCHF group (Figure 1a–d).

### 2.3. Serum TG and HDL

In this study, the HCHF diet caused a significant increase in serum TG levels in the HCHF (*p* = 0.003) and HCHF + KH (*p* = 0.001) group at week 8 compared to baseline. In the HCHF group, the serum TG level continued to rise significantly at week 16 (*p* = 0.001) compared to the C group. KH supplementation decreased the serum TG level (0.79 ± 0.05 mmol/L, *p* = 0.002) compared to the C group at week 16. Meanwhile, serum HDL was not decreased in the HCHF and HCHF + KH group compared to the C group at week 8 (*p* = 0.174). However, at week 16, serum HDL level in the HCHF group was significantly decreased compared to the control group (*p* = 0.018). Similarly, KH supplementation for 8 weeks also saw a significant increase in serum HDL level for the HCHF + KH group (0.86 ± 0.06 mmol/L, *p* < 0.001) (Figure 2a,b).

### 2.4. Systolic and Diastolic Blood Pressure

In this study, the HCHF diet caused a significant rise in SBP and DBP in the HCHF (SBP, *p* < 0.001; DBP, *p* < 0.001) and HCHF + KH (SBP, *p* < 0.001; DBP, *p* = 0.001) group at week 8 compared to their respective baseline values. In the HCHF group, the SBP (*p* = 0.002) and DBP (*p* = 0.001) continued to rise at week 16 in comparison to the C group. Meanwhile, KH supplementation improved both SBP (*p* < 0.001) and DBP (*p* < 0.001) at week 16 compared to the HCHF group (Figure 3a,b). 

### 2.5. Fasting Blood Glucose and Oral Glucose Tolerance Test 

The FBG and OGTT results (area under the curve (AUC)) are shown in Figure 4a,b. In this study, the HCHF diet did not increase FBG and AUC of OGTT in the HCHF (FBG: *p* = 0.431, AUC OGTT: *p* = 0.759) and HCHF + KH (FBG: *p* = 0.968, AUC OGTT: *p* = 0.187) group after 8 weeks. Meanwhile, the HCHF + KH group showed a decreased AUC of OGTT in week 16 compared to week 8 (*p* = 0.001). However, the comparison between the groups was not significant (HCHF + KH vs. C, *p* = 0.569; HCHF + KH vs. HCHF, *p* = 0.176). 

### 2.6. Serum Tumour Necrosis Factor Alpha (TNF-α), Interleukin-1-Beta (IL-1β) and Leptin

At the end of the study, serum TNF-α (*p* = 0.531; Figure 5a) and IL-1β levels (*p* = 0.914; Figure 5b) did not show any significant difference among the three study groups. Serum leptin was elevated significantly in the HCHF group (*p* = 0.004 vs. the C group), but this change was prevented in the HCHF + KH group (*p* = 0.002 vs. the HCHF group; Figure 5c).

### 2.7. Serum Adiponectin and Corticosterone

At week 16, serum adiponectin was higher in the HCHF + KH group compared to the HCHF group (*p* = 0.041; Figure 6a). Meanwhile, the serum corticosterone in the HCHF + KH group was significantly lower compared to both the C *(p* = 0.001) and the HCHF group (*p* = 0.019; Figure 6b). 

### 2.8. 11-Beta-Hydroxysteroid Dehydrogenase Type-1 (11βHSD1) Enzyme and Fatty Acid Synthase Enzyme (FASN)

At week 16, 11βHSD1 level in adipose tissue was significantly lower in the HCHF + KH group compared to the HCHF group (*p* = 0.017). No significant difference was observed in the 11βHSD1 level between the HCHF + KH and the C group (*p* = 0.268) (Figure 7a). Meanwhile, no significant differences were detected in the FASN level among the study groups (*p* = 0.301) (Figure 7b). 

### 2.9. Histomorphometry of Adipose Tissue, Liver, Pancreas and Renal Tissue

#### 2.9.1. Adipose Tissue

Histological examination of the adipose tissue showed that the HCHF diet caused adipocytes hypertrophy (Figure 8b), which was reflected by an increased area (*p* = 0.001) and perimeter of the adipocytes (*p* = 0.001) compared to the C group. In contrast, KH supplementation reduced these parameters significantly in the HCHF + KH group compared to the C group (*p* < 0.05) Figure 8c–e).

#### 2.9.2. Liver

Histological examination of the liver parenchyma showed that the hepatocytes of the HCHF group were larger compared to the C group (Figure 9a,b). The enlargement of hepatocytes was caused by the deposition of lipid droplets in their cytoplasm. Subsequent compression of the liver sinusoids was noted. Hepatocytes of the HCHF + KH group retained a normal morphology similar to the HCHF group (Figure 9c). 

#### 2.9.3. Pancreas

Histological examination of the pancreas revealed that the pancreatic islets of the HCHF group were reduced in number compared to the C group (Figure 10a,b). There is also deposition of fat in the pancreatic parenchyma of the HCHF group. Meanwhile, in the HCHF + KH group the pancreatic islets were more numerous compared to the HCHF group (Figure 10c). This shows that the KH treatment for eight weeks can prevent the reduction in pancreatic islets caused by the HCHF diet.

#### 2.9.4. Kidney Tissue

Histological examination of the renal tissue revealed widening of Bowman’s space in the HCHF group compared to the C group (Figure 11(ai,bii)). Meanwhile, the glomeruli of the HCHF + KH group showed normal morphology without widening Bowman’s space compared to the HCHF group rats (Figure 11(ci,cii)). This shows that KH treatment for eight weeks might protect against glomerular changes compared to the HCHF group. 

## 3. Materials and Methods

### 3.1. Liquid Chromatography-Mass Spectrometry (LC-MS) Analysis of Kelulut Honey 

Chemical analysis of KH was performed using ACQUITY Ultra High-Performance Liquid Chromatography (UHPLC) I-Class system (Waters, Milford, MA, USA) consisting of a binary pump, a vacuum degasser, an auto-sampler and a column oven. Phenolic compounds were chromatographically separated using a column ACQUITY UPLC HSS T3 (100 mm × 2.1 mm × 1.8 μm) (Waters, Milford, MA, USA) maintained at 40 °C. A linear binary gradient of water (0.1% formic acid) and acetonitrile (mobile phase B) (Merck, Darmstadt, Germany) was used as mobile phase A and B, respectively. The mobile phase composition was changed during the run as follows: 0 min, 1% B; 0.5 min, 1% B; 16.00 min, 35% B; 18.00 min, 100% B; 20.00 min, 1% B. The flow rate was set to 0.6 mL/min and the injection volume was 1 μL. The UHPLC system was coupled to a Vion IMS QTOF hybrid mass spectrometer (Waters, Milford, MA, USA) equipped with a Lock Spray ion source. The ion source was operated in the negative electrospray ionisation (ESI) mode under the following specific conditions: capillary voltage, 1.50 kV; reference capillary voltage, 3.00 kV; source temperature, 120 °C; desolvation gas temperature, 550 °C; desolvation gas flow, 800 L/h, and cone gas flow, 50 L/h. Nitrogen (>99.5%) was employed as desolvation and cone gas. Data were acquired in high-definition MSE (HDMSE) mode in the range *m*/*z* 50–1500 at 0.1 s/scan. 

### 3.2. High-Carbohydrate, High-Fat Diet Preparation

The high-carbohydrate and high-fat (HCHF) diet was prepared by mixing 175 g fructose (d-(−)-Fructose) (Chemiz, Shah Alam, Malaysia), 395 g sweetened condensed milk (Fraser & Neave Holdings Bhd., Kuala Lumpur, Malaysia), 200 g ghee (QBB Sdn. Bhd., Petaling Jaya, Malaysia), 25 g Hubble, Mendel, and Wakeman salt mixture (MP Biomedicals, California, CA, USA), 155 g powdered rat chow (Gold Coin Feedmills (M) Sdn. Bhd., Selangor, Malaysia) and 50 g water, in addition to 25% fructose (Chemiz, Shah Alam, Malaysia) drinking water. Diet preparation was performed according to the method by Wong and colleagues [25].

### 3.3. Kelulut Honey Preparation

Raw KH was harvested from the stingless honeybee species, *Heterotrigona itama,* from a local honeybee farm (Gombak, Selangor, Malaysia). The sample was collected during October 2020. The honey was stored in a glass jar at 4 °C until further use. KH was diluted with distilled water at a 1:1 ratio upon administration.

### 3.4. Experimental Animals

This study has been approved by the Animal Ethics Committee (UKMAEC), Laboratory Animal Resources Unit, Faculty of Medicine, Universiti Kebangsaan Malaysia (UKM) with the approval number ANAT/FP/2020/FAIRUS AHMAD/23-SEPT./1126-OCT.-2020-SEPT-202. A total of 18 Wistar male rats (*n* = 6/group) weighing between 250–300 g were taken from the Laboratory Animal Resources Unit, UKM (Kuala Lumpur, Malaysia). During the study period, the rats were placed in the Animal Laboratory of the Department of Anatomy, Faculty of Medicine, UKM (Cheras, Malaysia) while the acclimatisation process was carried out during the first two weeks. The rats were placed in pairs in plastic cages. The laboratory environment was maintained at a temperature of 25 ± 3 °C, with good air ventilation with a 12 h light/dark cycle.

### 3.5. Study Design

The rats were randomly divided into three groups, namely the control (C group) receiving normal rat chow, the HCHF group receiving an HCHF diet, and the HCHF + KH group receiving an HCHF diet and later supplemented with KH. Rats in the C group were given a normal chow (Gold Coin, Malaysia) and tap water ad libitum for 16 weeks. The other two groups received an HCHF diet and drinking water with 25% fructose ad libitum for 16 weeks. The HCHF + KH group was supplemented with KH via oral gavage at a dose of 1.0 g/kg/day, which had been shown to reverse MetS in a study by Ramli et al. (2019) from the 8th week after HCHF diet initiation until the end of the study. Meanwhile, an equivolume of distilled water was given to both the C and HCHF group to mimic the gavage stress [23]. After 16 weeks of HCHF diet induction, the rats were sacrificed by decapitation and the organs were harvested for analysis. 

### 3.6. Metabolic Parameters

#### 3.6.1. Measurement of Body Weight, Abdominal Circumference and Body Mass Index

The body weight of the rats was measured by using a digital weighing scale (Nimbus^®^ Precision Balances, Adam Equipment, Buckinghamshire, UK). Abdominal circumference and body length were also measured using a standard measuring tape. Meanwhile, the body mass index (BMI) was calculated using the following equation: BMI = body weight (g)/length^2^ (cm^2^). Measurements were taken at baseline, 8th and 16th week of the study period.

#### 3.6.2. Fat Percentage Measurement

Dual-energy X-ray absorptiometry (DXA) scans were performed using Hologic Discovery Densitometer (Hologic QDR-1000 System, Hologic Inc., Massachusetts, MA, USA) with Small Animal Analysis Software. During the DXA scan procedure, rats were anaesthetized with ketamine, xylazine, tiletamine and zolazepam (KTX) mixture. A whole-body scan was performed at baseline, week 8, and 16 to provide measurements of fat percentage.

#### 3.6.3. Serum Fasting Triglyceride and High-Density Lipoprotein

Blood was obtained from the retroorbital vein of the anaesthetized rats. For serum extraction, the blood was left to clot at room temperature for 20 min and subsequently centrifuged at 4000 rpm for 30 min. Determination of fasting serum triglyceride (TG) and high-density lipoprotein (HDL) levels was done using the automated clinical chemistry system model Dimension^®^ Xpand^®^ Plus (Siemens AG, Munich, Germany). Serum TG and HDL levels are measured at baseline, week 8, and 16.

#### 3.6.4. Blood Pressure Measurement

Systolic (SBP) and diastolic blood pressure (DBP) were measured by using the CODA^®^ tail-cuff blood pressure system (Kent Scientific Corporation, Torrington, CT, USA). Before measurement, rats were acclimatised to the CODA^®^ tail-cuff blood pressure system for 10 min, while maintaining the rat’s tail temperature between 32 °C and 35 °C. Blood pressure readings were taken at baseline, week 8 and week 16.

#### 3.6.5. Fasting Blood Glucose and Oral Glucose Tolerance Test

All rats were fasted for 12 h before the fasting blood glucose (FBG) and oral glucose tolerance test (OGTT) was performed. For rats receiving the HCHF diet, the 25% fructose drinking water was replaced with tap water. After 12 h of fasting, capillary blood glucose readings were taken and calculated as readings at the 0th minute through the capillary blood collection method. Immediately after the reading at the 0th minute was recorded, 2 g/kg of 40% glucose solution [D (+)-glucose, ChemPur^®^ Systerm^®^, Classic Chemicals Sdn. Bhd., Selangor, Malaysia] was given via oral gavage. Subsequently, glucose readings were retrieved at 30, 60 and 120 min. The OGTT test was performed at baseline, week 8 and week 16. 

### 3.7. Inflammatory and Obesity Markers

#### 3.7.1. Serum Tumour Necrosis Factor Alpha (TNF-α), Interleukin-1-Beta (IL-1β) and Leptin

Serum TNF-α, IL-1β and leptin levels were determined by a multiplex bead analysis, RADPCMAG-82K Rat Adipocyte Magnetic Bead Panel (Millipore, Watford, UK). Each sample (25 µL) was incubated with antibody-coated captured beads for 2 h under agitation at room temperature. After washing, the beads were further incubated with biotin-labelled anti-human cytokine and chemokine antibodies for 60 min, followed by streptavidin-phycoerythrin incubation for 30 min. Finally, the beads were washed and analysed with the Magpix reader (Luminex Corp., Austin, TX, USA). The standard curve of known concentrations of recombinant human cytokines was used to convert the fluorescent unit to the cytokine concentration. The detection limit of TNF-α, IL-1β and leptin were 0.1 pg/mL, 0.9 pg/mL and 2.3 pg/mL, respectively. Data were stored and analysed using Xponent 4.2 software (Luminex Corp., Austin, TX, USA). 

#### 3.7.2. Serum Adiponectin and Corticosterone

Serum adiponectin and corticosterone levels were measured in week 16 using the enzyme-linked immunosorbent (ELISA) technique with RatADP/Acrp30 (Adiponectin) and Rat/Chicken CORT (Corticosterone) ELISA kit (Elabscience, Houston, TX, USA) following the manufacturer’s instructions. The optical density was measured via Multiskan™ GO Microplate Spectrophotometer (Thermo Scientific, MA, USA) at the wavelength of 450 nm.

#### 3.7.3. Adipose Tissue Homogenization

The adipose tissue samples collected (0.3 g) were placed in an Eppendorf tube containing four to five iron balls. A total of 1 mL of radioimmunoprecipitation assay (RIPA) lysis buffer, 10 µL of phenylmethylsulphonyl fluoride (PMSF) and 10 µL of sodium vanadate (Na3VO4) was added to the tube. Next, the tissue was homogenized using a Bioprep-24 homogenizer (Hangzhou Allsheng Instrument CO., Ltd., Hangzhou, China) with a setting of 6.0 m/s for 30 s. This process was repeated twice to ensure that adipose tissue was completely lysed. The sample was then cooled on ice for 30 min. The mixture was then centrifuged at 12,000 rpm for 15 min at 4 °C. Before each centrifugation, the fat layer (upper layer) was removed to obtain the supernatant. This process was repeated twice until no additional layer of fat remained.

#### 3.7.4. 11-Beta-Hydroxysteroid Dehydrogenase Type-1 (11βHSD1) Enzyme and Fatty Acid Synthase Enzyme (FASN)

Adipose tissue 11βHSD1 and FASN levels were measured in week 16 using enzyme-linked immunosorbent (ELISA) techniques with Rat 11βHSD1 (11 Beta Hydroxysteroid Dehydrogenase Type-1) and Rat FASN (Fatty Acid Synthase) ELISA kit (Elabscience, USA). The optical density was measured via Multiskan™ GO Microplate Spectrophotometer (Thermo Scientific, Massachusetts, MA, USA) at the wavelength of 450 nm.

### 3.8. Histomorphometry of Adipose Tissue, Liver, Pancreas and Renal Tissue

Rats were sacrificed by decapitation following anaesthesia by the end of week 16. A mid-ventral abdominal incision was performed. Tissue excisions from visceral fat, liver, pancreas, and kidney were obtained, and preserved in a 10% buffered formalin (Merck, Darmstadt, Germany). Subsequently, the tissue was processed, embedded in paraffin wax, and sectioned (6 µm thick) before staining with hematoxylin and eosin. The stained tissue section was mounted on a glass slide and examined under a light microscope with 200× total magnification. Photomicrographs of visceral adipose tissue, liver, pancreas, and kidney parenchyma were taken using a light microscope (Carl Zeis Primo Star, Zeiss, Oberkochen, Germany). The ImageJ software (National Health Institute, Bethesda, MD, USA) was used for histomorphometric analysis of the adipose tissue.

### 3.9. Statistical Analysis

All parameters were analysed using SPSS^®^ software version 26 (IBM, Armonk, NY, USA). The normality of the data was assessed using the Shapiro–Wilk test. Normally distributed data were analysed using mixed-design analysis of variance (ANOVA) with small effect analysis as the post hoc test and presented as mean ± standard error of the mean (SEM). Wilcoxon-rank sum test (for paired samples) was also performed to assess the difference between baseline versus week 8 and week 8 versus week 16 for the same group. A *p*-value < 0.05 indicates a statistically significant difference.

## 4. Discussion

In this study, KH profiling found various phenolic compounds consisting of phenolic acids, such as cnidimonal, epigallocatechin(4β,8)-gallocatechin and E-*p*-coumaric acid, and flavonoid groups, such as naringin, kaempferol derivatives and isorhamnetin derivatives. These bioactive compounds could be responsible for the biological activities of KH observed in this study. For instance, kaempferol has been shown to inhibit adipocyte hypertrophy [26] and liver steatosis [27], as well as preventing the apoptosis of pancreatic islet cells [28]. Isorhamnetin is reported to ameliorate metabolic derangements, adipocytes hypertrophy and liver steatosis in obese mice [29]. Naringin has been found to suppress metabolic features and improve the number of functional β cells in rats [30,31]. Epigallocatechin gallate (EGCG) could inhibit 11βHSD1 activity and reduce cortisol levels [32,33], improve metabolic features and insulin sensitivity in animals [34], and renal histology [35,36,37]. *p*-coumaric acid could inhibit insulin resistance and increase adiponectin levels [38,39] Therefore, it is likely that phenolic compounds contained within KH contribute to the inhibition of MetS risk demonstrated in this study.

In this study, HCHF diet feeding for 16 weeks resulted in MetS in rats. In week 8, the HCHF diet gave rise to MetS components, such as central obesity reflected by increased body weight, WC and body fat percentage; hypertriglyceridemia and hypertension (increased SBP and DBP). In week 16, MetS worsened with the emergence of additional components, such as increased BMI, and decreased HDL. Therefore, this study showed that HCHF diet feeding for 16 weeks promoted MetS development, whereby 3 out of 5 interim joint statement criteria have been achieved. The results of this study were similar to the study by Ramli et al. (2019), who reported that the HCHF diet after 16 weeks could induce MetS risk factors, such as central obesity, hypertriglyceridemia and high blood pressure [23]. Moreover, HCHF also resulted in degenerative changes in the liver, kidneys and pancreas. Adipocytes showed an increase in the size and parameters, indicating hypertrophy due to excessive calorie intake. Whilst dyslipidemia resulted in the accumulation of fat within the hepatocytes. Histopathological evaluation revealed widening of Bowman’s space in the renal corpuscles, which was probably due to an increased blood pressure.

High caloric intake exceeding the energy requirement of an individual can cause weight gain [40]. In this study, high caloric intake through the HCHF diet for 16 weeks caused an increase in body weight and BMI as well as adipocyte hypertrophy. These observations were in line with previous studies which show that consumption of the HCHF diet for 16 weeks causes adipocyte hypertrophy and ectopic fat accumulation in rats [23]. During excessive calorie intake, adipocytes undergo hypertrophy to store their surplus energy. This growth continues until a critical diameter for visceral adipocytes is reached. Beyond this value, adipocytes stimulate the generation of new adipocytes from the precursor cells (adipogenesis) [41]. A study showed that dietary lipids stimulate the proliferation of adipose tissue progenitor cells in juvenile mice [42]. Excess calories systemically increased the mitogenic insulin-like growth factor 1 (IGF-1) levels, which is a systemic growth-promoting factor, whereas palmitoleic acid enhanced the sensitivity of progenitors to IGF-1, resulting in synergistic stimulation of proliferation. Furthermore, the high fructose content within the HCHF diet also influences the increase in TG levels. This is because fructose can be converted to glycerol-3-phosphate by bypassing the glycolysis pathway (phosphofructokinase) to produce a substrate for fatty acid synthesis [43]. Therefore, in this study, there was an increase in serum TG levels in HCHF-induced MetS rats. The increase in TG level gave rise to the accumulation of fat in hepatocytes as seen in histological observations. Under physiological conditions, FFA is taken up from plasma by the liver through specific fatty acid translocase (CD36) and fatty acid transport protein (FATP) family receptors undergoes β-oxidation, lipid droplet formation or VLDL formation [44]. However, fat accumulation due to excess calorie intake induces lipolysis, causing abundant FFAs release into the blood circulation [45]. This results in increased fatty acid uptake, to which the liver responds by increasing β-oxidation or esterification of fatty acid into TG [44]. Hepatic accumulation of TG is either utilised for VLDL formation or stored as lipid droplets in hepatocytes which contributes to the phenotypic hallmark of fatty liver disease. Similar findings were also observed by Rafie et al. (2018) whereby an increase in TG levels is reflected by fat accumulation within hepatocytes [24]. A study by Panchal et al. (2011) also demonstrated the accumulation of ectopic fat in liver tissue in rats after consumption of the HCHF diet for 16 weeks [46]. 

Increased TG levels promote TG-rich VLDL production [8]. Excess TG in VLDL will be transferred to HDL to form TG-rich HDL. This TG-rich HDL is then cleared from circulation via renal clearance, causing a decrease in HDL levels in the blood [47]. Therefore, this study found that there was a decrease in serum HDL levels of HCHF-induced MetS rats in week 16. The results of the study by Wong et al. (2018) also found lower HDL levels in rats given the HCHF diet after 16 weeks [25]. 

In this study, the HCHF diet increase SBP and DBP of the rats. According to Wong et al. (2018), the excessive sodium content in the HCHF diet may activate renin-angiotensin-aldosterone system (RAAS) [25]. RAAS activation causes vasoconstriction, which in turn increases blood pressure. Previous studies also found an increase in SBP and DBP readings after HCHF diet feeding [23,46]. In addition, central obesity is also known to cause glomerular hyperfiltration [48]. In this study, glomerular hyperfiltration resulted in glomerular changes with an increase in Bowman’s space. A study by Erejuwa et al. (2020) also showed that HFD feeding in rats caused glomerular changes with the presence of focal aggregate inflammatory cells, tubular necrosis, and glomerular atrophy [49]. Excessive visceral fat distribution is linked to hypertension by several possible mechanisms involved in hormonal, inflammatory and endothelial alteration [50]. In an insulin resistant state, the presence of excessive visceral fat can stimulate reabsorption of sodium and urates at the tubular level [51]. Some studies also consider the possibility of leptin to predict the onset of hypertension [52]. A recent study showed that leptin acts on adrenocortical cells to increase CYP11β2 expression and directly activates aldosterone secretion which modulates hypertension in female mice [53,54]. In addition, leptin also elicits symphato-exitatory effects which ensues the activation of RAAS [51]. A study showed that leptin acts as an autocrine mediator of Ang II-induced cardiomyocyte hypertrophy in hypertensive LVH rats, in which the treatment of telmisartan improves the myocardial remodeling in rats by inhibiting RAAS activity and leptin levels [55].

In this study, the HCHF diet did not increase FBG and AUC of glucose in rats receiving the diet. However, previous studies have found that the HCHF diet caused only mild impairment of glucose tolerance or insulin resistance [46]. Pre-diabetes is marked by an increase in plasma glucose concentration above the normal range but less than clinical diabetic values [56]. One of the characteristics of pre-diabetes is insulin resistance [57]. Insulin resistance can also be detected in individuals with a normal glucose tolerance [58]. A study by Townsend et al. (2018) found that 2.3% of normoglycemic young adult subjects experienced insulin resistance [58]. Therefore, no significant impairment of glucose tolerance in FBG and OGTT readings in this study does not rule out the possibility of insulin resistance. However, a study conducted by Ramli et al. (2019) showed no difference in insulin activity in rats induced with a HCHF diet after 16 weeks [23]. The authors hypothesized that insulin resistance may still be in its early stages and may take more than 16 weeks to produce significant changes. In this study, histological observation found a diminishing number of pancreatic islet cells in MetS rats which reflects the deterioration of β cells. Previous studies have also found that the HFD diet causes the deterioration of pancreatic β cells in rats [59]. Diminishing β cells within the pancreatic islet results in insulin resistance [60]. In addition, in response to its deterioration, the β cells will further increase insulin secretion, which exacerbates insulin resistance [61].

Administration of 1.0 g/kg of KH for 8 weeks in HCHF diet-induced MetS rats successfully inhibited MetS-associated changes, such as dyslipidemia, high blood pressure and insulin resistance. The dose of KH given (1.0 g/kg/day) was based on the previous study by Ramli et al. (2019) which used a similar dose had shown positive result in reversing metabolic symptoms [23]. Furthermore, an in vivo study by Azam et al. (2022) demonstrated that KH supplementation at doses 0.5, 1.0, and 2.0 g/kg to rats for 4 weeks did not cause toxicity or mortality [62]. The author also implied that the medium lethal dose for daily consumption of KH was higher than 2.0 g/kg rats’ BW. Despite the lack of effects on body weight, WC, BMI and body fat percentage, KH significantly decreases the size and perimeter of adipocytes. Study by Ramli et al. (2019) also showed the inhibition of adipocyte hypertrophy after KH administration in rats [23]. The anti-adipogenic properties of KH may be contributed by the active components within. Isorhamnetin, which is a metabolite of the flavonoid group quercetin, is known for its anti-adipogenicity [63]. Studies carried out in vitro have found that incubation of preadipocyte 3T3-L1 with isorhamnetin saw inhibition in the adipogenesis at dose > 10µM [29,64]. These studies also demonstrated that isorhamnetin suppressed adipocyte differentiation by inhibition of peroxisome proliferator-activated receptor (PPARγ) which is a master regulator of adipogenesis. This will reduce fat deposition in ectopic tissues, such as liver, skeletal muscle and heart, as well as visceral adipose depots [65]. 

Meanwhile, KH supplementation also improved serum TG and HDL levels. This observation is similar to the study by Rafie et al. (2018), whereby KH supplementation for 6 weeks ameliorated serum TG and HDL levels in rats receiving a HFD diet [24]. Furthermore, the study also found reduced fat deposition within hepatocytes in KH-treated groups. This histological improvement may be due to the decrease in TG levels because it is the main source of fat deposition in liver tissue [66]. The presence of phenolic compounds in KH may play an important role in limiting TG levels and liver fat accumulation. Studies show that gallic acid, quercetin, and kaempferol are among the phenolic compounds that can reduce fat accumulation [67]. According to another study, kaempferol and quercetin compounds were found to be able to inhibit pancreatic lipase enzyme activity with subsequent reductions in TG, FFA and body weight [68]. Meanwhile, Yoon et al. (2021) have also found that 0.002% *p*-coumaric acid in the HFD diet could reduce the accumulation of fat in the hepatocyte of HFD-induced obese rats [69]. In the study, *p*-coumaric acid suppressed the expression of lipogenic enzymes such as FASN and acetyl-CoA carboxylase (ACC) with consequent hepatic fatty acid oxidation. Therefore, this study shows that KH supplementation could prevent dyslipidemia and the progression of NAFLD.

Similarly, there was also an improvement in blood pressure measurement in KH-treated groups. The results of this study are supported by Ramli et al. (2019), who also found a decrease in the SBP and DBP in rats after KH supplementation for 8 weeks [23]. The improvement in blood pressure measurement can be explained by the anti-inflammatory properties of the bioactive compound within honey. MetS is linked to an increase in reactive oxidative stress and oxidative markers which contributes to the impairment of vascular dilatation [46,70]. Meanwhile, KH supplementation in rats (4.6 g/kg) for 30 days saw a reduction in pro-inflammatory cytokine production such as TNF- α, IL-1β, IL-6 and IL-8 [71]. The presence of phenolic compounds, such as kaempferol and *p*-coumaric acid in honey, might contribute to its anti-inflammatory properties as both compounds are known to inhibit the expression of nuclear factor kappa-light-chain-enhancer of activated B cells protein which regulates the production of inflammatory markers [72,73]. In addition, this study also observed normal glomerular histology without an increase in Bowman’s space in the HCHF + KH group rats compared to the HCHF group. Previous studies have also found improvements in the renal histology of rats that received Nigerian honey supplementation for 16 weeks [49]. This indicates that honey supplementation could prevent glomerular damage, leading to the normalisation of blood pressure. 

However, KH treatment for 8 weeks did not alter FBG and OGTT of rats in the HCHF + KH groups. Interestingly, despite the sugar content within KH, this study found that KH did not increase FBG and OGTT AUC of glucose in rats receiving the HCHF diet. A study found that quercetin and isorhamnetin promotes glucose uptake by increasing glucose transporter type 4 (GLUT-4) translocation in skeletal muscle cells, thus demonstrating its advantage in preventing hyperglycemia [74]. In addition, histological observation showed that the administration of KH for 8 weeks prevented the diminishing of pancreatic islets. Previous studies have also found similar results in STZ-nicotinamide-induced diabetic rats treated with KH [18]. According to the study, the administration of KH for 28 days did not cause an increase in FBG in diabetic rats. Furthermore, the study found that the inhibition of apoptosis markers by KH prevented the reduction in pancreatic islet cells. Meanwhile, a study found that diabetic rats treated with naringin (0.1 g/kg) for 8 weeks saw an increase in β cells. According to the study, naringin stimulates the Forkhead box M1 (Fox M1) protein expression which is an important transcription factor for β-cell proliferation [31]. While an in vitro study found that kaempferol at 10 μM was able to prevent cellular apoptosis by downregulating caspase-3 activity of β cells in hyperglycemic pancreatic islet cells [28]. Thus, KH is beneficial in preventing the formation of insulin resistance.

For inflammatory markers, this study found that the HCHF did not cause an increase in the serum levels of cytokines TNFα and IL-1β. A study by Stȩpień et al. (2014) found that the highly sensitive C-reactive protein (hs-CRP) is a more sensitive marker associated with obesity than IL-6 and TNF-α [75]. Similarly, a study conducted in Brazil on elderly cohorts found similar trend, whereby IL-1β, TNF-α and IL-12 were not statistically associated with MetS [76]. Instead, the presence of MetS was associated with higher IL-6 and CPR levels. However, the precise mechanism of these findings is not fully explained in these studies. Meanwhile, KH supplementation for 8 weeks did not alter serum cytokine levels. Similar result was also seen in other studies using polyphenol-rich food such as blueberries in insulin-resistant obese men and women as well as grapes in MetS men [77,78]. The lack of effects of KH on these inflammatory markers could be because these rats did not suffer from high levels of inflammation.

11βHSD1 is an enzyme that catalyses the conversion of cortisone to corticosterone [79]. Studies have shown that the expression of the 11βHSD1 enzyme is high in obese individuals [80]. Therefore, increased activity of this enzyme promotes the production of corticosterone hormone [79]. Excessive production of corticosterone hormone, in turn, causes disturbances in physiological metabolism leading to central obesity, insulin resistance, dyslipidemia and hypertension [81]. In this study, the HCHF diet induced a higher expression of the 11βHSD1 enzyme, which might be responsible for the metabolic changes in the rats. The HCHF + KH group showed lower 11βHSD1 expression, which indicates a reduction in corticosterone hormone secretion, and the amelioration of MetS risk factors. The molecular mechanism of KH in inhibiting 11βHSD1 may be contributed by the presence of phenolic compounds contained within it. An in vitro study demonstrated that pre-treatment of rat liver microsomal vesicles with EGCG caused oxidation of luminal NADPH to NADP+ [82]. The redox shift results in the indirect inhibition of 11βHSD1 mediated cortisol production at the substrate level. Meanwhile, Pathak et al. (2017) also found a reduction in circulating corticosterone level and 11βHSD1 pancreatic activity in HFD-fed rats [34]. Additionally, the study also saw improvements in rat weight reduction, fat mass, glycemic control and insulin concentration after EGCG supplementation. 

A previous study found that hypertrophy and hyperplasia of adipocytes in obese rats is linked to high levels of corticosterone [81]. The corticosterone hormone is important in regulating adipogenesis and the secretion of adipokines, such as leptin and adiponectin [83]. Therefore, in this study, the inhibition of the 11βHSD1 enzyme and corticosterone hormone production by KH could inhibit adipocyte hypertrophy, reduce leptin levels, and increase adiponectin levels in HCHF + KH rats. In turn, inhibition of adipocyte hypertrophy as well as regulation of hormone secretion prevents the formation of central obesity.

The increase in adiponectin levels is closely related to the decrease in TG levels. Studies have found that adiponectin could activate 5’ adenosine monophosphate-activated protein kinase (AMPK) [84]. The activation of AMPK further causes phosphorylation of acetyl CoA carboxylase (ACC), which promotes fat burning in the tissue [85]. This, in turn, reduces the production of TG and VLDL, as well as increases the HDL levels in the blood. Concurrently, the reduction in hepatosteatosis as observed in the liver histology of this study also reflects suppression of TG production. 

In addition, increasing adiponectin levels can also increase insulin sensitivity through the phosphorylation of insulin receptor substrate 1 (IRS-1) and p3 MAPK [86]. Phosphorylation of these proteins leads to the membrane translocation of GLUT-4 [85]. This process in turn causes the uptake of glucose by body tissues. In addition, an in vitro study also found that adiponectin hormone protects pancreatic β cells from undergoing apoptosis [87]. Therefore, in this study, increasing adiponectin levels can prevent insulin resistance by increasing insulin sensitivity as well as protecting pancreatic islets from apoptosis due to MetS.

Leptin is closely related to increased blood pressure activity by activating sympathetic activity [88]. Studies have found that the activation of renal sympathetic activity caused by leptin triggers RAAS activation, which leads to hypertension [89]. The RAAS activation subsequently causes vascular vasoconstriction [90]. Therefore, in this study, the reduction in leptin by KH was followed by the normalisation of blood pressure, which in turn restores glomerular deterioration due to hyperfiltration, as seen in histological observations. 

On the other hand, this study showed that there was no difference in FASN enzyme levels between all the study groups. In contrast, other studies have associated the increase in FASN expression with insulin resistance in metabolic syndrome rat model [91,92]. These studies suggested that the increase in FASN activity is prompted by hyperglycemia caused by insulin resistance which triggers the activation of a de novo lipogenic enzyme [92,93]. Suzuki et al. (2015) postulated that these changes are brought about by increased methylation and acetylation of histone in the promoter-enhancer region of FASN gene in the setting of hyperglycemia [92]. Therefore, the lack of increased FASN activity in this study might be due to the unelevated glycemic measurement observed in rats. Meanwhile, the bioactive component within KH did not modify the FASN parameter. An in vitro study conducted by Gómez-Zorita et al. (2017) observed a reduction of FASN expression in pre-adipocytes cultured with flavonoid compound apigenin and hesperidin at 25 µM for 8 days [94]. However, no significant reduction in FASN expression was observed with kaempferol at 25 µM. This suggests that phenolic compounds within KH do not have any effect on FASN enzyme levels. A summary of the mechanism of KH in reversing metabolic changes in MetS rats is depicted in Figure 12.

The importance of this study is to examine the action of KH in treating MetS. Therefore, this study will form the basis for KH supplementation for MetS patients. However, the administration of the HCHF diet in the current study was only successful in influencing three out of five components of MetS, thus did not fully reflect the full potential of KH in reversing MetS. Therefore, some recommendations in further studies include increasing the duration of the HCHF diet beyond 16 weeks to increase the probability of influencing all components of MetS as contained in the JIS criteria. In addition, the physiochemical parameters of KH should also be incorporated, as they determine the quality of honey, which makes it beneficial. According to a study by Kek at al. (2017), the KH produced by Heterotrigona itama collected at different time demonstrated variations in physicochemical and antioxidant properties [95]. The KH that was used in this experiment was harvested in October 2020. Collecting KH samples at different times may yield different result. Furthermore, although the amelioration of MetS by KH supplementation that was seen in this study may be contributed by its rich phenolic compound; however, the molecular mechanism of these active compounds in reversing MetS is not fully understood. Therefore, a more in-depth study on the mechanism of these phenolic compounds may help in understanding the beneficial effect of KH on MetS. 

## 5. Conclusions

This study shows that the administration of KH for 8 weeks to rats with HCHF diet-induced MetS can improve central obesity, dyslipidemia, and hypertension. The inhibition of these MetS components is also indicated by reduced levels of 11βHSD1, serum corticosterone and leptin, as well as increased serum adiponectin level. Furthermore, histological observations suggest KH administration inhibits adipocyte hypertrophy, fat accumulation in hepatocytes, protects pancreatic islets from deterioration, and prevents glomerular changes in the renal tissue. The phenolic compounds in KH could be responsible for inhibiting metabolic changes in rats with MetS. However, the mechanisms of inhibition by KH phenolic compounds need to be proven in further studies.

## Figures and Tables

**Figure 1 molecules-28-02790-f001:**
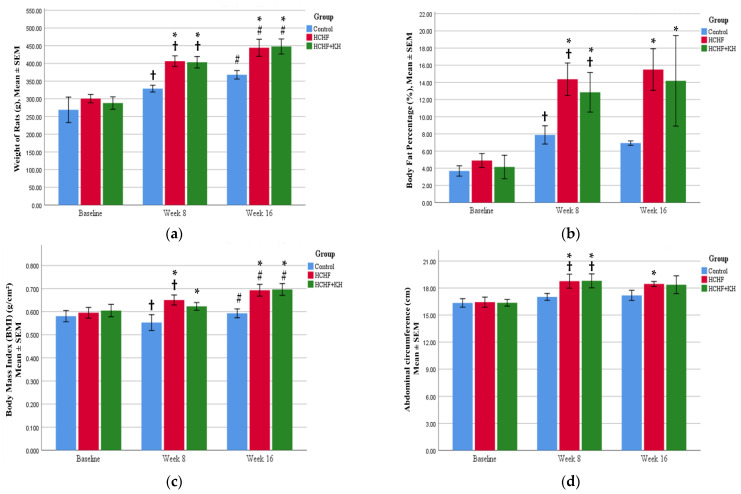
Bar charts showing changes in the (**a**) body weight, (**b**) body fat percentage, (**c**) body mass index and (**d**) abdominal circumference of rats in group the Control, HCHF and HCHF + KH. † Significant difference within the same group compared to baseline, *p* < 0.05, # Significant difference within the same group at week 16 compared to week 8, *p* < 0.05, * Significant difference compared to control within the same week, *p* < 0.05. The error bars represent the standard error of the mean with *n* = 6 in each group. Abbreviation: HCHF (HCHF diet-induced MetS), HCHF + KH (MetS group receiving KH).

**Figure 2 molecules-28-02790-f002:**
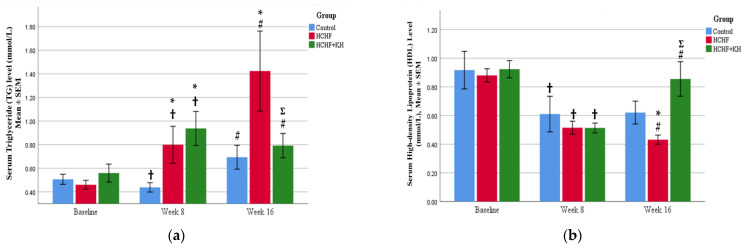
The bar chart showing changes in the serum (**a**) TG and (**b**) HDL levels of rats in group Control, HCHF and HCHF + KH. † Significant difference within the same group compared to baseline, *p* < 0.05, # Significant difference within the same group at week 16 compared to week 8, *p* < 0.05, * Significant difference compared to the Control within the same week, *p* < 0.05, Ʃ Significant difference compared to HCHF within the same week. The error bars represent the standard error of the mean with *n* = 6 in each group. Abbreviation: HCHF (HCHF diet-induced MetS), HCHF + KH (MetS group receiving KH).

**Figure 3 molecules-28-02790-f003:**
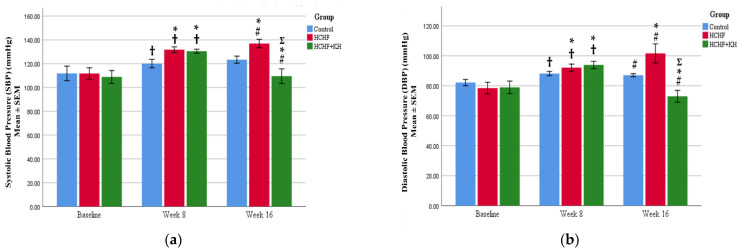
Bar charts showing changes in the (**a**) SBP and (**b**) DBP blood pressure of rats in group Control, HCHF and HCHF + KH. † Significant difference within the same group compared to baseline, *p* < 0.05, # Significant difference within the same group at week 16 compared to week 8, *p* < 0.05, * Significant difference compared to the Control within the same week, *p* < 0.05, Ʃ Significant difference compared to HCHF within the same week. The error bars represent the standard error of the mean with *n* = 6 in each group. Abbreviation: HCHF (HCHF diet-induced MetS), HCHF + KH (MetS group receiving KH).

**Figure 4 molecules-28-02790-f004:**
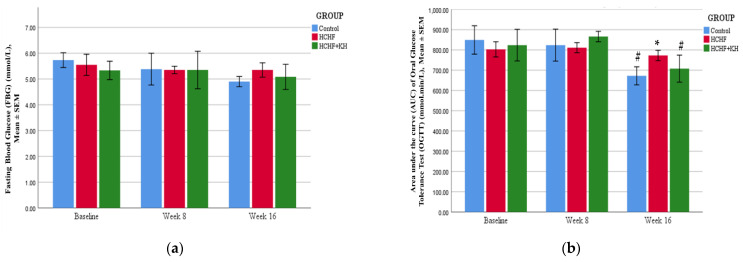
The bar chart showing changes in the (**a**) fasting blood glucose (FBG) and (**b**) area under the curve (AUC) of oral glucose tolerance test (OGTT) of rats in group Control, HCHF and HCHF + KH. # Significant difference within the same group at week 16 compared to week 8, *p* < 0.05, * Significant difference compared to the Control within the same week, *p* < 0.05. The error bars represent the standard error of the mean with *n* = 6 in each group. Abbreviation: HCHF (HCHF diet-induced MetS), HCHF + KH (MetS group receiving KH).

**Figure 5 molecules-28-02790-f005:**
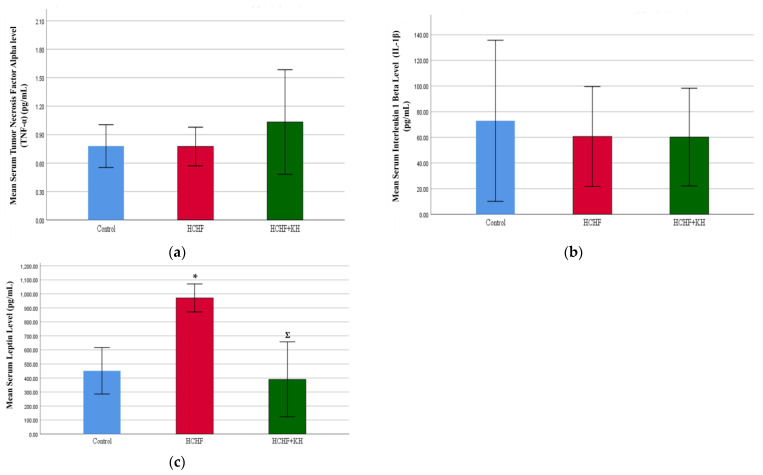
Bar chart showing measurement of serum (**a**) Tumour Necrosis Factor Alpha (TNF-α), (**b**) Interleukin-1-Beta (IL-1β) and (**c**) Leptin in rats from group Control, HCHF and HCHF + KH at week 16. * Significant difference compared to the Control, *p* < 0.05, Ʃ Significant difference compared to HCHF. The error bars represent the standard error of the mean with *n* = 6 in each group. Abbreviation: HCHF (HCHF diet-induced MetS), HCHF + KH (MetS group receiving KH).

**Figure 6 molecules-28-02790-f006:**
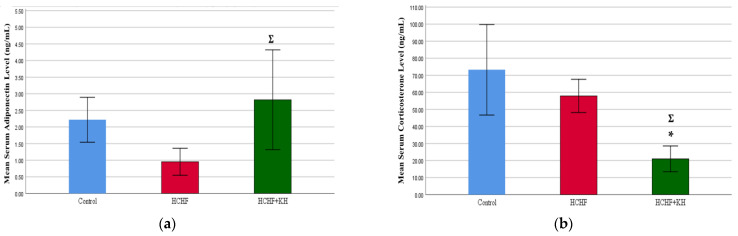
The bar charts show the serum levels of (**a**) adiponectin and (**b**) corticosterone in rats from group Control, HCHF and HCHF + KH at week 16. * Significant difference compared to the Control, *p* < 0.05, Ʃ Significant difference compared to HCHF. The error bars represent the standard error of the mean with *n* = 6 in each group. Abbreviation: HCHF (HCHF diet-induced MetS), HCHF + KH (MetS group receiving KH).

**Figure 7 molecules-28-02790-f007:**
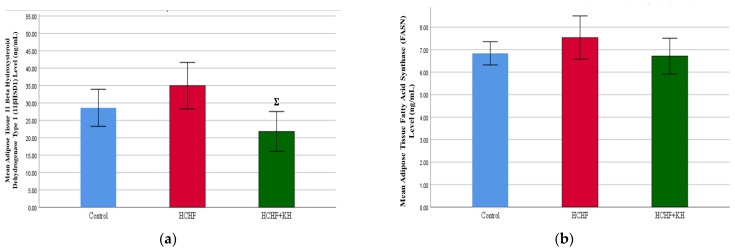
Bar charts showing measurement of (**a**) 11 beta-hydroxysteroid dehydrogenase type-1 (11βHSD1) and (**b**) fatty acid synthase enzyme (FASN) of adipose tissue in rats from group Control, HCHF and HCHF + KH at week 16. Ʃ Significant difference compared to HCHF. The error bars represent the standard error of the mean with *n* = 6 in each group. Abbreviation: HCHF (HCHF diet-induced MetS), HCHF + KH (MetS group receiving KH).

**Figure 8 molecules-28-02790-f008:**
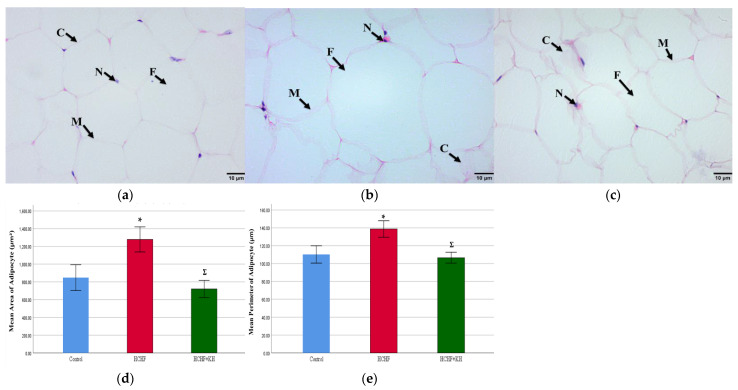
Photomicrograph of adipose tissue in (**a**) Control, (**b**) HCHF, and (**c**) HCHF + KH group at 400× total magnification. The size of adipocytes was indicated by measurement of (**d**) area and (**e**) perimeter. * Significant difference compared to the Control, *p* < 0.05. Σ Significant difference compared to HCHF, *p* < 0.05. Abbreviation: C—cytoplasm, F—fat storage, N—nucleus and M—plasma membrane. The error bars represent the standard error of the mean with *n* = 6 in each group. Abbreviation: HCHF (HCHF diet-induced MetS), HCHF + KH (MetS group receiving KH).

**Figure 9 molecules-28-02790-f009:**
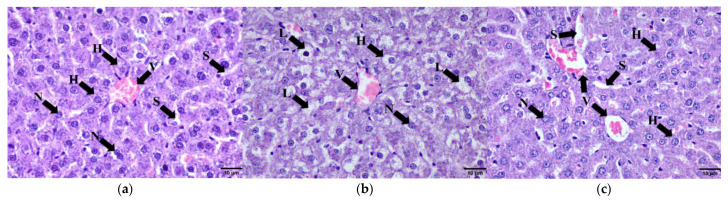
Photomicrograph of hepatocytes in (**a**) Control, (**b**) HCHF, and (**c**) HCHF + KH group at 400× total magnification. Abbreviations: H hepatocytes, L—lipid droplets, N—nucleus, S—liver sinusoids, V—central vein.

**Figure 10 molecules-28-02790-f010:**
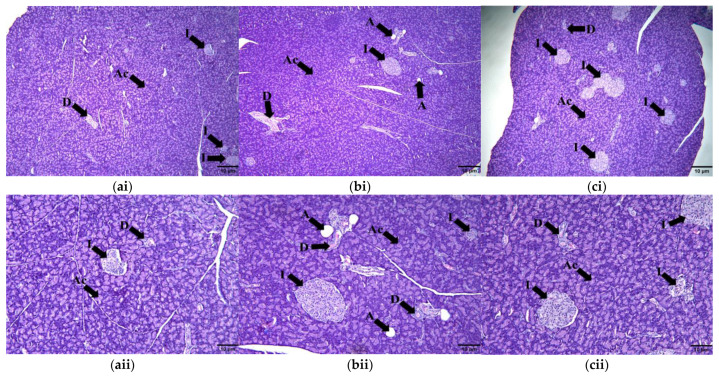
Photomicrographs of pancreas in (**ai**) Control, (**bi**) HCHF, and (**ci**) HCHF + KH group at 40× total magnification and (**aii**) Control, (**bii**) HCHF, and (**cii**) HCHF + KH group at 100× total magnification. Abbreviations: I—islet of Langerhans, Ac—acinus, D—duct, A—adipocyte.

**Figure 11 molecules-28-02790-f011:**
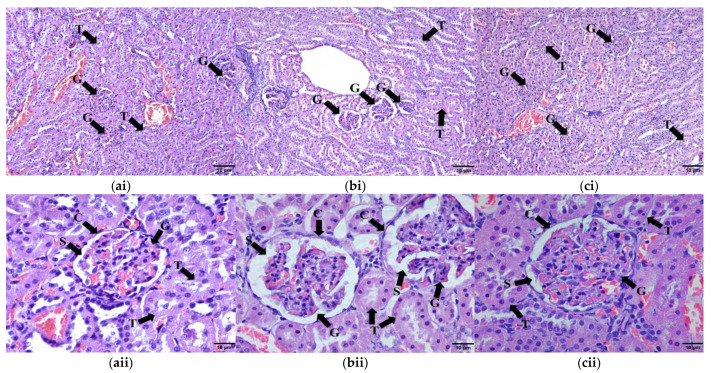
Photomicrographs of renal tissue in (**ai**) Control, (**bi**) HCHF, and (**ci**) HCHF + KH group at 100× total magnification and (**aii**) Control, (**bii**) HCHF, and (**cii**) 410 HCHF + KH group at 400× total magnification. Abbreviations: G—glomerulus, S—Bowman’s space, C—parietal layer of the Bowman’s capsule and T—proximal and distal tubules.

**Figure 12 molecules-28-02790-f012:**
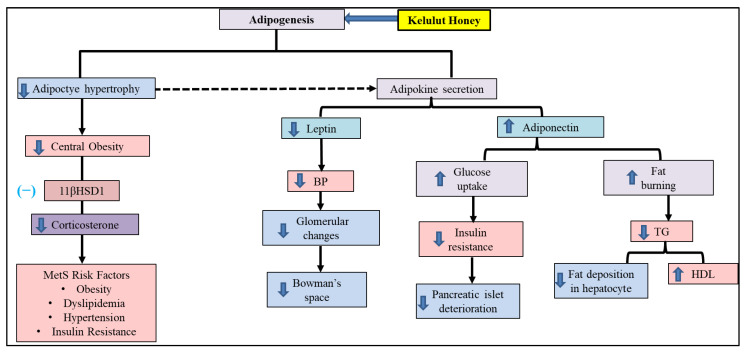
Association between KH treatment and metabolic changes of MetS risk factor. The HCHF diet causes adipocyte hypertrophy, which is closely related to increased leptin hormone secretion, reduced adiponectin hormone and elevated 11βHSD1 enzyme activation (which catalyses corticosterone hormone secretion). Increased secretion of hormones such as leptin and corticosterone and inhibition of adiponectin hormones lead to the formation of MetS risk factors. The administration of KH reverses metabolic changes by inhibiting 11βHSD1 enzyme activity, which suppresses excess corticosterone secretion. KH also promotes changes in adipokine secretion, i.e., reduced leptin hormone and increased adiponectin hormone. All these effects contribute to the inhibition of MetS. Blue arrow: Increase/decrease due to KH.

**Table 1 molecules-28-02790-t001:** Result of LCMS profiling of Kelulut honey.

No.	Mass (Da)	Compound Name	Formula
1	404.0896	Cnidimonal	C23H16O7
2	610.13226	Epigallocatechin(4β,8)-gallocatechin	C30H26O14
3	678.15847	1,3,5-*O*-Tricaffeoyl-quinic acid	C34H30O15
4	368.11073	5-Feruloylquinic acid	C17H20O9
5	164.04734	E-*p*-coumaric acid	C9H8O3
6	368.11073	Methyl chlorogenate	C17H20O9
7	594.15847	Kaempferol-3-*O*-neohesperidoside	C27H30O15
8	594.15847	Kaempferol-3-*O*-rutinoside	C27H30O15
9	580.17921	Naringin	C27H32O14
10	768.21129	Viscumneoside VII	C34H40O20
11	770.22694	Isorhamnetin-3-*O*-(2G-α-l-rhamnosyl)-rutinoside	C34H42O20
12	450.11621	Luteolin-7-*O*-α-d-glucoside	C21H22O11
13	450.11621	Neoastilbin	C21H22O11
14	576.12678	Procyanidin A2	C30H24O12
15	468.10565	Apocynin B	C24H20O10
16	610.15338	Kaempferol-3,7-di-*O*-β-d-glucoside	C27H30O16
17	624.16903	Isorhamnetin-3-*O*-β-rutinoside	C28H32O16

## Data Availability

Data is available.

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
