# Peer review of "The Mechanism of Kelulut Honey in Reversing Metabolic Changes in Rats Fed with High-Carbohydrate High-Fat Diet"

_molecules, 2023, doi:10.3390/molecules28062790_

Round 1

Reviewer 1 Report

I found study interesting; following points may further improve the manuscript.

1.   Why a painful method (retro orbital) was used for drawing blood? Was the method was approved by Animal Ethics Committee? What was the survival rate of rat after taking out blood?

2.    Figure 5 and others, Error bar is found to be very high.

3.    KH was diluted with distilled water at a 1:1 ratio, mention the reason of dilution.

4.   On what basis KH dose of 1.0 g/kg/day (what was volume per serve) was decided. Is there any toxicity data available with KH administration, if yes mention in discussion part?

5.    The composition of honey changes with the seasons and flowering plants availability. How does this study correlate with all these factors? Collection time (season/months) should be highlighted?

6.      Physicochemical parameters of KH should be incorporated somewhere.

Author Response

Thank you for your feedback. Please see the attachment for the point-by-point response.

Reviewer 2 Report

Line 130: Anthropometrical Measurement - this must be changed to the correct form : anthropometrical is from ancient greek 'human' and refers to the measurement of the human individual

Line 138: Body Composition Measurement - this must be changed to the correct form: only fat percentage was measured.

Line 215: The adipose tissue, liver, pancreas and renal tissue were then dissected - I propose to change to: tissue excisions from visceral fat, liver, pancreas and kidney were obtained

Line 219: viewed - I propose to change to: examined

Line 219: Photomicrographs of adipocytes, hepatocytes, pancreatic islets and renal glomerulus were taken using a light microscope (Carl Zeis Primo Star, Zeiss, Germany) while ImageJ software (National Health Institute, USA) was used to analyse the histomorphometry of the adipose tissue. - I propose to change to: Photomicrographs of visceral adipose tissue, liver, pancreas and kidney parenchyma were taken using a light microscope (Carl Zeis Primo Star, Zeiss, Germany). The ImageJ software (National Health Institute, USA) was used for histomorphometric analysis of the adipose tissue.

Line 366: Histological examination of the adipose tissue showed that the HCHF diet caused adipocyte hypertrophy (Figure 8a–c), which was reflected by an increased area (p = 0.001) and

perimeter of the adipocytes (p = 0.001) compared to the C group. In contrast, KH supple-

mentation reduced these parameters significantly in the HCHF+KH group compared to

the C group (p < 0.05; Figure 8d–e). - I propose to change to: Histological examination of the adipose tissue showed that the HCHF diet caused adipocytes hypertrophy (Figure 8b), which was reflected by an increased area (p = 0.001) and perimeter of the adipocytes (p = 0.001) compared to the C group. In contrast, KH supplementation reduced these parameters significantly in the HCHF+KH group compared to the C group (p < 0.05) Figures (8c–e).

Line 372: Photomicrograph of adipose tissues in (a) Control, (b) HCHF, and (c) HCHF+KH group

under a light microscope with 400x total magnification. - I propose to change to: Photomicrograph of adipose tissue in (a) Control, (b) HCHF, and (c) HCHF+KH group at 400x total magnification.

- (photomicrograph itself means a digital image taken through a microscope).

Line 380: Histological examination of the liver showed that the hepatocytes of the HCHF group appeared larger in size compared to the C group (Figure 9a-b). In addition, fat accumulation was noted to accumulate within the hepatocytes, causing the cells to be compressed thus making the liver sinusoid of the HCHF group not being visible. Meanwhile, the hepatocytes of the HCHF+KH group retained a normal morphology similar to the HCHF group (Figure 9c). - I propose to change to: Histological examination of the liver parenchyma showed that the hepatocytes of the HCHF group were larger compared to the C group (Figure 9a-b). The enlargement of hepatocytes was caused by the deposition of lipid droplets in their cytoplasm. Subsequent compression of the liver sinusoids was noted. Hepatocytes of the HCHF+KH group retained a normal morphology similar to the HCHF group (Figure 9c).

Line 387: - I propose to: delete under a light microscope, at 400x total magnification.

Abbreviation: H; yellow arrow (hepatocyte), F and L; green arrow (fat storage and lipid droplet), N; blue arrow (nucleus), S; red arrow (sinusoid) and 389 V; black arrow (central vein). - I propose to change to: Abbreviations: H - hepatocytes, L - lipid droplets, N - nucleus, S - liver sinusoids, V - central vein.

-          (there is no need for color differentiation of the arrows when the tissue structures are clearly marked with letters - it is rather distracting)

-          (is there any difference between the fat storage – F and lipid droplet - L?)

-          (there is missing label on the 8b V – central vein)

Line 392: Histological examination of the pancreas found that the pancreatic islet of the HCHF

group was reduced in number compared to the C group [ Figure 10a-b (i. and ii.)]. Meanwhile, the HCHF+KH group appeared to have more pancreatic islets compared to the

HCHF group rats (Figure 10c (i and ii)). This shows that the KH treatment for eight weeks

can prevent the reduction of pancreatic islets compared to the HCHF. - I propose to change to: Histological examination of the pancreas revealed that the pancreatic islets of the HCHF

group were reduced in number compared to the C group (Figure 10a-b). Meanwhile, in the HCHF+KH group the pancreatic islets were  more numerous compared to the HCHF group (Figure 10c). This shows that the KH treatment for eight weeks can prevent the reduction of pancreatic islets caused by the HCHF diet.

Line 397: Figure 10. Photomicrograph of pancreatic islets in (ai) Control, (bi) HCHF, and (ci) HCHF+KH  group under a light microscope with 40x total magnification and (aii) Control, (bii) HCHF, and (cii) HCHF+KH group under a light microscope with 100x total magnification. Abbreviation: I; red arrow (islet of Langerhans), A; yellow arrow (acinar cell), D; black arrow (interlobular duct).

- I propose to change to: Figure 10. Photomicrographs of pancreas in (ai) Control, (bi) HCHF, and (ci) HCHF+KH group at 40× total magnification and (aii) Control, (bii) HCHF, and (cii) HCHF+KH group at 100× total magnification. Abbreviations: I - islet of Langerhans, A - acinus, D - duct.

-          (there is no need for color differentiation of the arrows when the tissue structures are clearly marked with letters - it is rather distracting)

-      in the photomicrographs bi, bii I would also mark the deposition of fat in the pancreatic parenchyma A – adipocytes

Line 401: - I propose to change to: Kidney

Line 402: - I propose to change to:  Histological examination of the renal tissue revealed widening

Line 409: - I propose to change to: Photomicrographs of renal tissue in (ai) Control, (bi) HCHF, and (ci) HCHF+KH group at 100× total magnification and (aii) Control, (bii) HCHF, and (cii) 410 HCHF+KH group at 400× total magnification. Abbreviations: G - glomerulus, S - Bowman’s space, C - parietal layer of the Bowman’s capsule and T - proximal and distal tubules.

-          (there is no need for color differentiation of the arrows when the tissue structures are clearly marked with letters - it is rather distracting)

-          (it would be appropriate to unify the size of the arrows as well as the size of the font in the photos)

Line 438: - I propose to change to: Moreover, HCHF also resulted in degenerative changes in the liver, kidneys and pancreas.

Line 441: - I propose to change to: Histolopathological evaluation revealed widening of Bowman's space in the renal corpuscles, which was probably due to an increased blood pressure.

Line 445: increase in body weight, BMI and adipocyte – adipocyte? (unclear)

Line 480: renin–angiotensin system (RAAS) - - I propose to change to: renin-angiotensin-aldosterone system (RAAS)

Line 682: The inhibition 682 of these MetS components results is also indicated by reduced adipose tissue – unclear

-          (the discussion is very extensive, I propose to shorten it)

Author Response

(The authors gave the same response as above.)
